# The Impact of an Automation System Built with Jenkins on the Efficiency of Container-Based System Deployment

**DOI:** 10.3390/s24186002

**Published:** 2024-09-16

**Authors:** Giwoo Hyun, Jiwon Oak, Donghoon Kim, Kunwoo Kim

**Affiliations:** 1Grida Tech Inc., Busan 48520, Republic of Korea; gi020602@gridatech.com (G.H.); kdhmail2084@gridatech.com (D.K.); kimiron4@gmail.com (K.K.); 2Department of Nursing, TongMyong University, Busan 48520, Republic of Korea

**Keywords:** container, server virtualization, cloud computing, continuous integration, continuous deployment

## Abstract

This paper evaluated deployment efficiency by comparing manual deployment with automated deployment through a CI/CD pipeline using Jenkins. This study involved moving from a manual deployment process to an automated system using Jenkins and experimenting with both deployment methods in a real-world environment. The results showed that the automated deployment system significantly reduced the deployment time compared to manual deployment and significantly reduced the error rate. Manual deployment required human intervention at each step, making it time-consuming and prone to mistakes, while automated deployment using Jenkins automated each step to ensure consistency and maximized time efficiency through parallel processing. Automated testing verified the stability of the code before deployment, minimizing errors. This study demonstrates the effectiveness of adopting a CI/CD pipeline and shows that automated systems can provide high efficiency in real-world production environments. It also highlights the importance of security measures to prevent sensitive information leakage during CI/CD, suggesting the use of secrecy management tools and environment variables and limiting access rights. This research will contribute to exploring the applicability of CI/CD pipelines in different environments and, in doing so, validate the universality of automated systems.

## 1. Introduction

Fast and reliable deployments are essential in modern software development environments. Traditional deployment methods lead to long development cycles, frequent deployment failures, and inefficient cross-team communication. DevOps and CI/CD were introduced to address these issues.

DevOps is a portmanteau of Development and Operations and aims to foster collaboration between software development and operations teams to speed up development and make systems run more reliably. DevOps uses automation, continuous monitoring, collaboration tools, and more to shorten development cycles, improve the quality of software, and ensure that everything from development to deployment is seamless [1]. This enables organizations to deliver shorter development cycles and better-quality software.

Continuous integration (CI) is the process by which developers frequently merge code changes and use automated tests to ensure that the changes integrate well with existing code. This process plays an important role in shortening the overall development cycle by enabling the early detection and correction of problems in the early stages of development.

Continuous delivery (CD) is the process of running code integrated through CI through an automated build system and conducting testing to make it ready for deployment to production. This ensures that it is stable and ready to be deployed at any time, significantly reducing the time to market for your software.

Continuous feedback is the process of monitoring the performance and stability of deployed software, making continuous improvements based on user feedback and system logs. This allows you to quickly identify and resolve issues in production, increasing overall system reliability.

CI/CD is an important component of DevOps, which stands for continuous integration and continuous delivery. CI is the process by which developers frequently integrate code changes and validate them through automated build and test systems to find problems early [2]. CD is a set of processes that automatically pushes this integrated code into the deployment environment, ensuring automation and consistency in the deployment process. This process encourages developers to frequently commit code changes, which are then automatically integrated and tested to increase the reliability and efficiency of the deployment process. This makes DevOps and CI/CD a powerful tool to increase collaboration between teams and improve both the quality and speed of software deployment (Figure 1).

Containers provide an isolated environment that contains an application and all the dependencies needed to run that application, ensuring the portability and consistency of the application [3]. Container technologies such as Docker are a prime example of providing such environments. Containers provide operating system-level virtualization, making it easy to move applications to different environments and ensuring consistency in the execution environment. Containers also help you run more applications simultaneously by using resources efficiently. CI/CD automates the software development and deployment process to save time and labor, reduces the likelihood of errors through automated testing and deployment, and ensures that code changes can be deployed quickly and delivered to users quickly [4]. It was introduced to address the long development cycles, frequent deployment failures, and ineffective cross-team communication that occurred with traditional software deployment methods [5]. As such, CI/CD is becoming a standard and increasingly necessary part of the modern software development environment.

Jenkins is an open-source automation server that is widely used as a tool to automate continuous integration and continuous delivery during software development and deployment. Jenkins’ functionality can be extended through a variety of plugins, allowing you to build CI/CD pipelines that support different languages and technology stacks. For example, Jenkins can support build automation, test automation, and deployment automation, all through different plugins. This means that every time a developer commits a code change, Jenkins automatically runs the build, test, and deployment process, increasing the reliability and efficiency of the deployment process. Supported by an active open-source community, Jenkins is an important tool for implementing such automation systems [6]. In addition, Jenkins’ plugin architecture gives developers the flexibility to extend its functionality and build customized pipelines as needed (See Figure 2 and Figure 3).

The process of deploying a container-based system consists of the following steps: creating an image, storing the image, and running the container. First, you build your application’s code into a Docker image. In this step, you create a Dockerfile that contains the source code of your application and the necessary dependencies and then create a Docker image based on it [8]. A Dockerfile is a file that defines your application’s execution environment in code, which allows you to run your application in a consistent environment. The created image is uploaded to an image repository, such as Docker Hub. In this process, you use an image repository to store and manage images [9]. Image repositories support versioning, allowing you to use specific versions of images as needed. Finally, you run the containers based on the stored images. This process can be automated using container orchestration tools, which can make the deployment and management of containers more efficient. For example, orchestration tools like Kubernetes allow you to effectively manage and deploy hundreds of containers [10]. These automation tools can go a long way toward reducing operational costs and increasing the reliability of your system.

The process of containerizing a backend service consists of the following steps: preparing the code, creating a Dockerfile, building the image, and deploying the image through a CI/CD pipeline. First, the service’s code is prepared. In this step, you check the source code and the required dependencies and create a Dockerfile based on them. After you create the Dockerfile, you use it to build a Docker image. The built image is uploaded to an image repository and automatically deployed through a CI/CD pipeline. By building a CI/CD pipeline, you can increase the reliability and efficiency of your deployments by having code changes automatically tested, built, and deployed. The pipeline also allows you to detect and fix errors that may occur during the deployment process early on.

This study analyzes the impact of an automated system using Jenkins on the efficiency of container-based system deployment. By building a CI/CD pipeline with Jenkins and automating the deployment process of backend services through it, we aimed to reduce deployment time, reduce error occurrences, and increase overall system deployment efficiency. This will contribute to demonstrating the benefits of an automated deployment system and explore the applicability of CI/CD pipelines in different environments.

## 2. Materials and Methods

In this study, we aimed to improve deployment efficiency by converting the existing manual deployment process to a CI/CD pipeline using Jenkins. To do so, we first analyzed the existing manual deployment process in detail.

### 2.1. Analysis of Manual Deployment Process

The manual deployment process consists of the following steps. First, the developer makes code changes in the local environment and commits them to the Git repository. Next, they connect to the deployment server via SSH and download the latest code using Git. They check the dependency libraries required by the downloaded code, update them if necessary, and restart the application using the system service manager. Finally, they manually test and verify that the deployed application is working properly. The problem with this manual deployment process is that it is time-consuming, labor-intensive, and prone to errors due to human intervention. For example, there are a variety of potential problems, such as errors during the dependency installation phase due to compatibility issues with certain library versions or connectivity issues during the server connection phase. These issues make deployment times unpredictable and make it difficult to respond quickly in an emergency.

Some tasks reduce developer productivity and reduce the time available for critical development tasks. The process of manually installing dependencies and restarting applications is particularly time-consuming, which can lengthen deployment cycles and delay the time it takes for new features or fixes to reach users. To address these inefficiencies, an automated deployment system is needed.

### 2.2. Automation of Manual Deployment Processes

After analyzing the manual deployment process, we found several steps that could be automated. First, we were able to automate the steps of automatically detecting code changes to start the pipeline, downloading and building code, running automated tests, deploying images, restarting services, and completing final testing and verification. Based on these steps, we configured a CI/CD pipeline in Jenkins to build an automated process.

The first step in the CI/CD pipeline is for Jenkins to detect code changes. This is detected in real time via webhooks in the Git repository, and the pipeline is automatically started when a code change is detected. This automatically starts the deployment process whenever a developer commits a code change, ensuring that deployments happen quickly and without manual intervention.

In the second step, you download the latest code and build the application image using Docker. Docker allows you to build your application in a consistent environment, which minimizes environment-specific build errors. This helps reduce the number of unexpected issues that can arise during deployment.

In the third step, you run the automated tests set up in your Jenkins pipeline to verify the stability of your code. These tests include unit tests, integration tests, system tests, and more, and can quickly discover and fix errors that occur during each test phase. Automated tests play an important role in ensuring the quality of your code before deployment, which helps you deliver stable software to your users.

The fourth step is to upload the built image to Docker Hub, pull the latest image from the deployment server, and run it. This process is fully automated, and communication between Docker Hub and the deployment server ensures that the latest image is always available.

In the fifth step, you restart the containers with an automated script. This script runs on the deployment server and runs the container with a new image, replacing the existing instance. This ensures that deployed containers are updated to the latest version and also makes it easy to roll back if needed. Finally, additional automated tests set up in the Jenkins pipeline verify that the deployed containers are working properly. This allows you to validate that your containers are working reliably even after deployment. In this course, for added security, we made sure that all sensitive information used by the CI/CD pipeline (API keys, database credentials, etc.) is managed securely through a secrets management tool and utilized environment variables to enhance security.

To quantitatively evaluate the efficiency of this automation process, we calculated the deployment time reduction rate (*DTRR*) and error reduction rate (*ERR*) between manual and automated deployments.

The deployment time reduction ratio (*DTRR*) is the difference in deployment time between manual and automated deployments, expressed as a percentage, to show the time saved due to automation:$$DTRR=\frac{Manual\;deployment\;time-Automatic\;deployment\;time}{Manual\;deployment\;time}\times 100$$

The error rate reduction (ERR) is a numerical measure of the effectiveness of automation, calculated as a percentage reduction in error rates between manual and automated deployments:$$ERR=\frac{Number\;of\;manual\;deployment\;errors-Number\;of\;auto-deployment\;errors}{Number\;of\;manual\;deployment\;errors}\times 100$$

With these two formulas, we could clearly evaluate the performance of the automation system and visually see the improvements in terms of deployment time and errors compared to manual deployments. This quantitative analysis can help one make a case for the benefits of an automated system more clearly.

### 2.3. Experimental Backend System Configuration

In this experiment, we assumed that we were updating a backend system that was being serviced by an arbitrary cloud. For this experiment, the cloud environment was Azure’s App Service Plan B1 version, a commercial cloud service provided by Microsoft, and the random backend system was an educational performance management system used by Gridatech in Busan, Republic of Korea, a Korean virtual reality nursing education developer. This performance management backend system provides services through API requests and provides various functions such as searching the student grade database, entering grades, modifying grades, and authenticating users. It is also a service that is frequently updated to meet the changing education market.

### 2.4. Deployment Process of Existing Backend System

Figure 4 shows what a traditional deployment of a performance management backend system might look like.

The manual deployment process for a performance management backend system works like this. First, the developer develops the application in a local environment and commits code changes to the Git repository. On the test server, they clone and pull the code from the Git repository, stop the live service, and update the service code. Once the updates are complete, the test server restarts the service and tests it for proper operation. After testing is complete, the same procedure is repeated on the live server. On the live server, the code is cloned and imported from the Git repository, the live service is stopped, and the service code is updated. Then, the service is restarted on the live server and its functionality is verified. This manual deployment process requires developers to perform each step manually, which is time-consuming and error-prone due to human intervention. In particular, it is inefficient due to the many repetitive tasks such as stopping and restarting the service and updating the code, making it difficult to respond quickly in an emergency.

### 2.5. Change to Automated Deployment Structure

To address these challenges, we leveraged containers and Jenkins to automate our deployment process. Figure 5 shows how we did it.

The automated deployment process for a performance management backend system works like this. A developer develops the application in a local environment and commits code changes to a Git repository. The Git system sends a notification to the Jenkins server via a webhook, which the Jenkins server detects. The Jenkins server then starts building the container image. Once the build is complete, the Jenkins server tests the image. If the tests are passed, it SSHs to the live server, updates it with the new container image, and recreates the container using that image. Finally, it restarts the container on the live server to complete the deployment. This automated process allows developers to drastically reduce the time spent on deployment tasks and minimizes the chance of deployment errors.

### 2.6. Hardware Specifications for Experiments

The hardware specifications of the development PC we prepared to perform the experiments in this paper were as follows.

CPU: 12th Gen Intel^®^ Core™ i7-12700H (Santa Clara, CA, USA)GPU: Nvidia GeForce RTX 3080 Ti Laptop (Santa Clara, CA, USA)RAM: Samsung DDR5-5600 64GB (Suwon-si, Gyeonggi-do, Republic of Korea)SSD: Micron 3400 MTFDKBA1T0TFH (Boise, ID, USA)

Jenkins, Docker, and the Live/Test server ran on a single server, and we assumed that they ran on their respective hardware or in the cloud. The hardware specifications for a server fulfilling these roles are as follows.

CPU: Intel(R) Xeon(R) Silver 4208 × 2ea (Santa Clara, CA, USA)RAM: Samsung DDR5-5600 128GB (Suwon-si, Gyeonggi-do, Republic of Korea)SSD: DELL PERC H750 adp0 (DISK 6ea raid 0) (Round Rock, TX, USA)

### 2.7. Experimental Conditions

Finally, in this paper, we set up the following experimental conditions to compare the time difference between the two proposed deployment processes.

Suppose you have a single administrator managing multiple performance management backend systems that use the same source code, and you need to distribute updates to multiple servers.To do this, admins run manual and automated deployments 1, 5, 15, and 30 times and then measure and compare the time they take.Then, experiments are run with the same hardware on a local network to reduce build time errors due to network latency, hardware performance differences between deployments, etc.

## 3. Results

In this study, we evaluated deployment efficiency by comparing manual deployment with automated deployment through a CI/CD pipeline using Jenkins. The results showed that the automated deployment system significantly reduced deployment time compared to manual deployment and significantly reduced the error rate.

### 3.1. Overview of the Experimental Environment

The automated deployment process significantly reduced deployment times compared to manual deployments. In our experiment, we measured deployment times by running each deployment process 1, 5, 15, and 30 times. The results are shown in Table 1 and Figure 6 below.

The deployment time reduction rate (*DTRR*), mentioned in Section 2.2, provides a numerical assessment of how time-efficient automated deployments are. The *DTRR* represents the difference in deployment time between manual and automated deployments as a percentage and is calculated using the following formula:1 deployment: $DTRR=\frac{14-3}{14}\times 100\approx 78.57\%$5 deployments: $DTRR=\frac{59-3}{59}\times 100\approx 94.92\%$15 deployments: $DTRR=\frac{184-4}{184}\times 100\approx 97.83\%$30 deployments: $DTRR=\frac{367-10}{367}\times 100\approx 97.27\%$

These results show that the automated deployment system provides significant time savings, especially for repetitive deployments. The *DTRR* of around 79-98% is a good indication of the efficiency of the automated deployment system.

### 3.2. Error Analysis in Deployment Processes

Errors that can occur in manual deployment can occur at various stages. These include compatibility issues when manually installing or updating dependency libraries, network errors when pulling the latest code from a Git repository, or configuration errors when restarting a service. These errors often occur when human intervention is required, and the deployment process can be delayed or fail due to developer mistakes or unexpected system conditions.

Errors that can occur in automated deployment are often caused by errors in automated scripts or pipeline settings. These include a bug in a Jenkins pipeline script, an error in building a Docker image, or a failed automated test of the deployed application. These errors can usually be resolved with a configuration or script fix and can be detected and responded to quickly with automated monitoring and alerting systems.

We measured the number of errors that occurred in manual and automated deployments to evaluate how much the automated deployment process can reduce the number of errors (Table 2).

We evaluated the effectiveness of error reduction between the manual and automated deployments by calculating the error reduction rate (*ERR*), mentioned in Section 2.2. The *ERR* is the difference in the number of errors between manual and automatic deployments, expressed as a percentage, and is calculated using the following formula.

The ERR results for each number of deployments are shown below:1 deployment: $ERR=\frac{0-0}{0}\times 100=0\%$5 deployments: $ERR=\frac{2-0}{2}\times 100=100\%$15 deployments: $ERR=\frac{4-0}{4}\times 100=100\%$30 deployments: $ERR=\frac{7-1}{7}\times 100\approx 85.71\%$

The ERR results show that automated deployments can significantly reduce errors compared to manual deployments. The high ERR values (over 85%) emphasize that automation is an important factor in significantly improving the reliability of the deployment process.

### 3.3. Summary of Results

The results of this study clearly show that automated CI/CD pipelines provide significant benefits in terms of time and errors compared to manual deployments. Automated deployments were able to reduce deployment times by approximately 79–98% and error rates were reduced by up to 100%. These results illustrate why DevOps and CI/CD tools are essential in modern software development and deployment environments.

## 4. Discussion

In this study, we evaluated deployment efficiency by comparing manual deployment with automated deployment through a CI/CD pipeline using Jenkins. Our results showed that automated deployment systems have significant advantages over manual deployment in several ways. Below are the key findings of this study and our discussion.

During automated deployments, deployment times were significantly reduced compared to manual deployments, especially for iterative deployments. Manual deployments were time-consuming because they required human intervention to perform each step, whereas automated deployments maximized time efficiency because CI/CD tools like Jenkins handled all steps automatically. Our experiments showed that automated deployments can reduce deployment time by up to 98% compared to manual deployments, which illustrates why DevOps and CI/CD tools are essential in modern software development and deployment environments. Automated deployments are also more efficient because they can be processed concurrently on the server, allowing multiple tasks to be performed in parallel.

The automated deployment process eliminates the need for human intervention, significantly reducing the number of mistakes and errors that can occur in manual deployments. Thanks to automated testing and a consistent deployment environment, we were able to thoroughly validate the stability of the code before deployment. This allowed us to proactively detect and fix any errors that could occur after deployment, resulting in reliable software delivery. In particular, errors in automated deployments were reduced by more than 85% compared to manual deployments, highlighting the importance of automated deployment systems in significantly improving the reliability of the deployment process. It also reduced the fatigue and mistakes that could occur in repetitive tasks, increasing the overall reliability of the deployment process.

By adopting an automated deployment system, developers are freed from repetitive and tedious tasks, allowing them to focus on more important work. With manual deployments, long wait times and repetitive tasks distracted developers and left them with less time for important development tasks. Automated systems, on the other hand, streamline and automate the deployment process, allowing developers to focus on solving more important problems without having to worry about the deployment process. This contributes to the overall productivity of the development team, which can then respond quickly to emergencies and deliver features and fixes to users faster. This is in line with the results of reduced deployment time and fewer errors presented in this study.

One of the important considerations when adopting an automated deployment system is information security. It is essential to take appropriate security measures to ensure that sensitive information used in the CI/CD pipeline, such as API keys, database credentials, environment variables, and other sensitive data, is not leaked. To this end, we suggest the following practices. First, use secrets management tools such as HashiCorp Vault, AWS Secrets Manager, and Azure Key Vault to securely store and manage sensitive information [11]. Second, manage sensitive information by setting it as environment variables rather than embedding it directly in the code. CI/CD tools like Jenkins provide the ability to securely handle environment variables. Third, apply the principle of least privilege, ensuring that each user and system component has only the minimum privileges they need. In Jenkins, you can leverage role-based access control (RBAC) to fine-grain permissions. Fourth, monitor the logs generated during the deployment process and detect anomalies so that you can respond quickly. Also, be careful to ensure that logs do not contain sensitive information. Fifth, harden the security settings of tools like Jenkins and update them regularly to remove known vulnerabilities. For example, check the security settings of plugins and disable those you do not need.

As this study was conducted on a specific backend service, further research is needed on different types of services and more complex systems [12,13]. In particular, it is important to apply the CI/CD pipeline in different environments to validate the generality of the automation system. For example, it is necessary to evaluate the performance in environments with limited server resources or in different cloud service environments to see how adaptable the automation tool is in practice. It is also necessary to study potential problems or security issues that may arise during the automation process. Future research should focus on improving the ease of use and security of automation tools and broadening the scope of automated testing. This will allow us to explore the applicability of CI/CD in different environments and provide a more concrete picture of how much benefit an automated system can actually provide. Furthermore, through continuous monitoring and improvement, we can further increase the reliability and stability of CI/CD systems.

Through this study, we found that by building a CI/CD pipeline utilizing Jenkins, we were able to overcome the inefficiencies of the manual deployment process and significantly improve deployment efficiency. We successfully achieved our goals of reducing deployment time and reducing error rates, which shows that automated systems can provide great efficiency in real-world operations. We also recognized the importance of information security and proposed ways to prevent information leakage during CI/CD through appropriate security measures. In future research, it will be important to explore the applicability of CI/CD in various environments to verify the universality of automated systems.

## Figures and Tables

**Figure 1 sensors-24-06002-f001:**
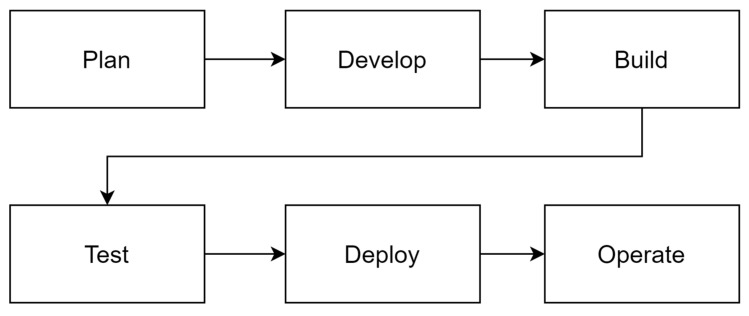
CI/CD deployment process.

**Figure 2 sensors-24-06002-f002:**
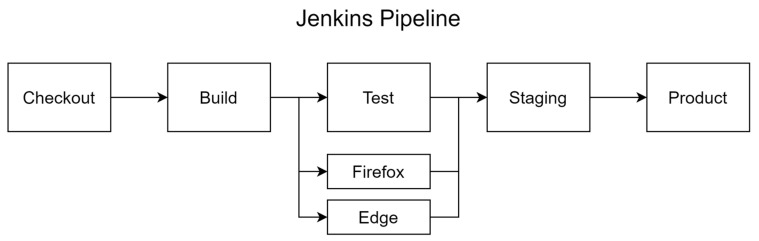
Jenkins pipeline.

**Figure 3 sensors-24-06002-f003:**
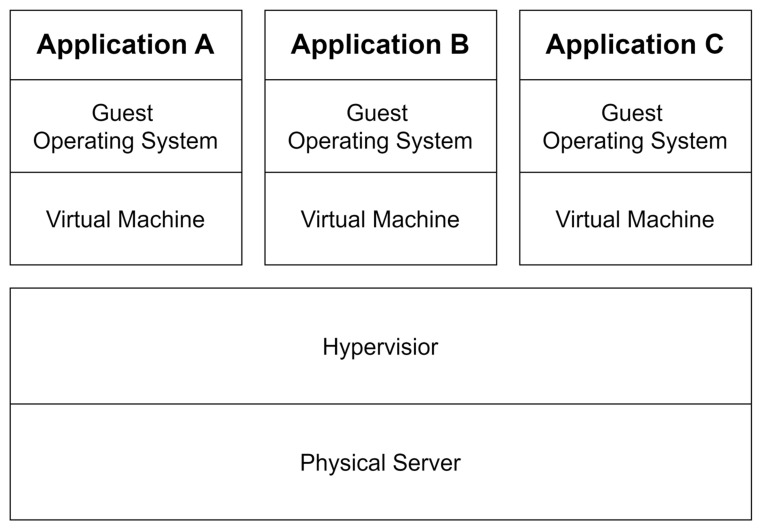
The concept of containers [7].

**Figure 4 sensors-24-06002-f004:**
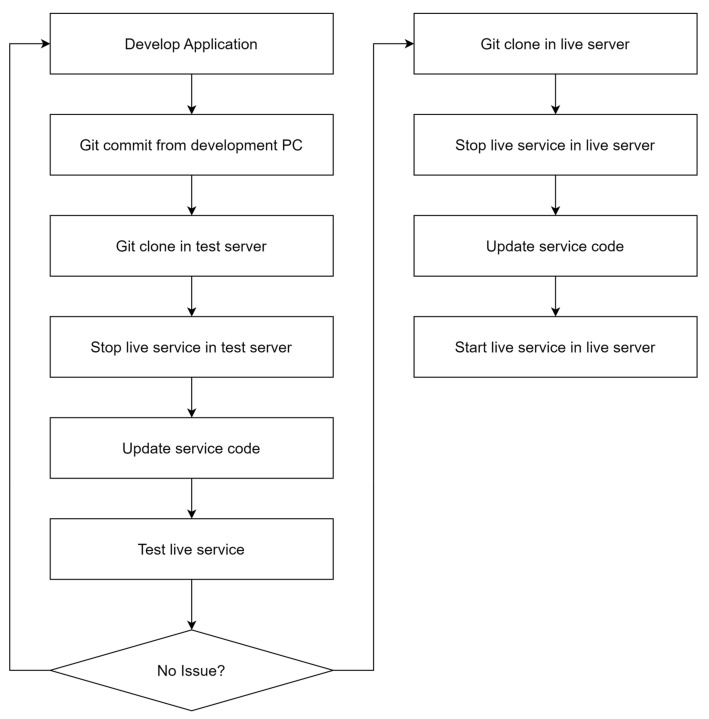
Existing performance management backend system deployment process.

**Figure 5 sensors-24-06002-f005:**
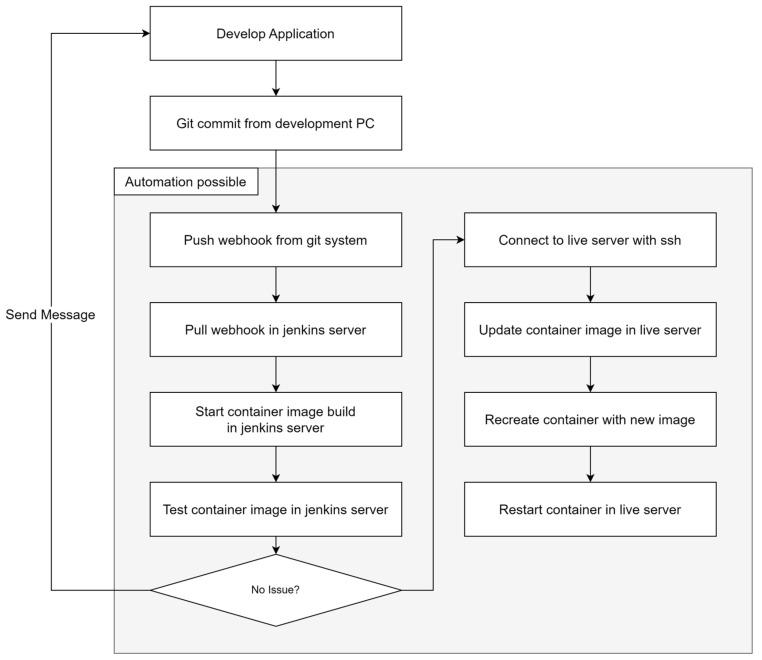
Deployment process with automation.

**Figure 6 sensors-24-06002-f006:**
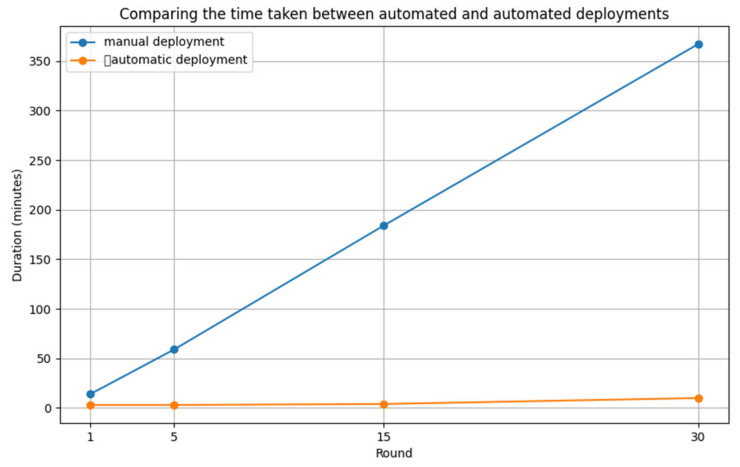
Time measurement result graph.

**Table 1 sensors-24-06002-t001:** Timing results for manual and automated deployments.

Count	Manual Deployment	Automated Deployment
1	00:14	00:03
5	00:59	00:03
15	03:04	00:04
30	06:07	00:10

**Table 2 sensors-24-06002-t002:** Error results for manual and automated deployments.

Count	Number of Manual Deployment Errors	Number of Auto-Deployment Errors
1	0	0
5	2	0
15	4	0
30	7	1

## Data Availability

The source code and datasets used in this study are the property of Gridatech in Busan, Republic of Korea and are therefore subject to restrictions.

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
