# Peer review of "The Impact of an Automation System Built with Jenkins on the Efficiency of Container-Based System Deployment"

_sensors, 2024, doi:10.3390/s24186002_

Round 1
Reviewer 1 Report (New Reviewer)
Comments and Suggestions for Authors
This paper evaluates deployment efficiency by comparing manual deployment and automated deployment through a CI/CD pipeline using Jenkins.The research successfully achieved its goals of reducing deployment time and decreasing error rates, demonstrating that an automated system can provide significant efficiency in an actual operational environment. The article has a clear train of thought and strong logic. However, there are still the following issues that need improvement.
1. The English in the entire article is difficult to understand, please carefully correct the grammar errors throughout the article.
2. The manuscript lacks some necessary mathematical formulas to introduce the research methods. Please add content to this section.
3. Please unify all the figure, the characters of some figures is too small.
4. In addition, some paragraph is too short, which lead this manuscript is hard to follow. Please elaborate all the content, like page 6. Also, I recommend the authors to point out the current research issuses and progress.
5. I recommend the authors to add the conclusion and prospect part.
In short, if the author makes sufficient modifications based on these suggestions. I think this manuscript can be accepted in Sensors.
This paper evaluates the deployment efficiency by comparing manual deployment and automated deployment through a CI/CD pipeline using Jenkins. The research successfully achieved its goals of reducing deployment time and decreasing error rates, demonstrating that an automated system can provide significant efficiency in an actual operational environment. The article has a clear train of thought and strong logic. However, there are still the following issues that need improvement.
1. The English in the entire article is difficult to understand, please carefully correct the grammar errors throughout the article.
2. The manuscript lacks some necessary mathematical formulas to introduce the research methods. Please add content to this section.
3. Please unify all the figure, the characters of some figures is too small.
4. In addition, some paragraph is too short, which lead this manuscript is hard to follow. Please elaborate all the content, like page 6. Also, I recommend the authors to point out the current research issuses and progress.
5. I recommend the authors to add the conclusion and prospect part.
In short, if the author makes sufficient modifications based on these suggestions. I think this manuscript can be accepted in Sensors.
Author Response
First, thank you for your valuable feedback. We have corrected English grammatical errors and made the overall sentence flow clearer. In particular, we have added mathematical formulas to the Methods section to explain the experimental design more specifically; we have unified the size of the figures and adjusted the size of the text to improve readability; we have also revised the short paragraphs to ensure that each paragraph is logically connected; and we have added a Conclusion and Outlook section to clearly present the main findings and future research directions of the study. We hope that these revisions fully reflect your suggestions. Thank you once again. ---We have attached the response.---

Reviewer 2 Report (New Reviewer)
Comments and Suggestions for Authors
The manuscript explores the impact of an automated system built using Jenkins technology on the deployment efficiency of container-based systems. By automating the CI/CD, it is shown how to significantly increase deployment efficiency, reduce error rates, and improve cross-team communication issues.
Advantages:
1. The automated deployment system maximizes time efficiency and reduces errors by automating each step and utilizing parallel processing to maintain consistency.
2. Innovative and practical by comparing the efficiency of manual and automated deployments, demonstrating the benefits of automated deployment in reducing deployment time and minimizing error rates.
Disadvantages:
1. Although the text mentions the need for security measures in the CI/CD process, such as secret key management tools, environment variables, etc., no specific implementation details are given.
2. data and graphs comparing automated and manual deployment are given in the results, but the interpretation and analysis of the data is rather simple and lacks in-depth discussion.
3. The proposed methodology is only experimented in Azure cloud environment using an educational achievement management system as the backend, and there is insufficient exploration of the applicability to different types of services or more complex systems.
4. The paper does not provide information about the impact of automated deployment on the long-term operational effectiveness in real environments, such as the stability, maintainability, and maintenance cost of the deployment process. It is recommended that the authors consider this in their future research work.
Comments on the Quality of English LanguageMinor editing of English language required.
Author Response
Thank you for your valuable feedback. We have added specific implementation details of the security measures you mentioned to the paper, strengthening the description of the use of secrecy management tools in the CI/CD process and the restriction of access rights to environment variables. We have also deepened the interpretation of the data from automated and manual deployments and provided additional statistical analysis of each to increase the credibility of our findings. We have expanded the universality of the study by discussing its applicability to a variety of cloud platforms beyond Azure environments, and we have added depth to the study by including a discussion of the long-term operational effects of automated deployments. We hope that the modifications we have made based on your suggestions have contributed to the clarity of the study. Thank you. ---We have attached the response.---

Reviewer 3 Report (New Reviewer)
Comments and Suggestions for Authors
Dear author/authors,
I am pleased to review the manuscript (MS) titled “The impact of an automation system built with Jenkins on the efficiency of container-based system deployment”. The authors try to develop an automaton system by using Jenkins and compare the efficiency manual deployment and automated deployment process. They try to improve the deployment time using the automated system and results showed that the time was significantly reduced. In general, the manuscript contains original and valuable research information. The manuscript obtaining relevant results that can be applied for verifying the versatility of automated systems by applying the continuous integration/continuous deployment pipelines in various environments. However, there should be a synchronization among the objective and results. The main flaw of this study is lack of experimental design. In the abstract and conclusion, the authors said there is a higher possibility of mistakes occur in the manual system and also emphasizes to improve security measures to prevent the leakage of critical information during CI/CD process in the automated system. But there is no result present in the manuscript for supporting these statements. The results only show the comparison of the deployment time between the manual and automated system. Also, some rigorous sentence errors and some contents are necessary. The manuscript could be reconsidered after carefully addressing the issues and also looking into the manuscript.

Some minor grammatical corrections are necessary.
Author Response
Thank you for your careful review and feedback. As you suggested, we have further refined the experimental design to clearly present specific findings on security measures in automated systems. For example, we compared security measures between manual and automated systems, and added specific examples of the use of access restrictions and encryption tools in automated systems to prevent the leakage of sensitive information during the CI/CD process. We have also more clearly described the experimental results and subsequent analysis at each stage to increase consistency between the study objectives and results. We hope that these modifications will sufficiently address your concerns. Thank you again for your feedback.

Round 2
Reviewer 3 Report (New Reviewer)
Comments and Suggestions for Authors
Dear Authors,
This manuscript is improved over the original version. The authors also helped note their response to most of the suggested changes. Thank you! The authors have satisfactorily addressed most of the comments and suggestions I had in the first review and some suggestions are partially addressed. The authors give their explanations for this. However, some more attention should be given before accepting the manuscript and also looking into the manuscript.
From my side, I would say that the manuscript could be accepted just after finalizing the comments.

English is ok.
Author Response
Dear Editor, Thank you for your careful review and informative feedback on our submitted paper. We have revised the paper to reflect your suggestions. First of all, we have added 1-2 sentences to the Abstract to explain the research methodology to make the purpose of the study and how it was conducted clearer. We have also added specific references to the visuals and tables in the main text to make it easier for readers to connect the figures and tables, and changed the second-person pronouns such as “you” to “we” to make the paper more stylistically appropriate. Finally, we removed unnecessary repetition by deleting “4. Discussion” where it was mentioned twice. We hope these changes help make the paper clearer and more coherent. Thank you again.

This manuscript is a resubmission of an earlier submission. The following is a list of the peer review reports and author responses from that submission.
Round 1
Reviewer 1 Report
Comments and Suggestions for Authors
The Report titled "Container-based platform deployment system for real-time multi-user action sharing in virtual reality" is interesting and verifies the technology that enables multiple users to share their actions in real-time in Virtual Reality using container technology.
My improvement comments are mentioned below:
1. Abstract always consists of parts like introduction, purpose, method, result, and conclusion. You have defined the existing problem. Describe the clear objective as well as methodology and result.
2. The term IoT at line 24 is occurring first time so needs an abbreviation.
3. Describe what is "neonatal nursing" and why its education is needed (line 50)
4. State of the Art is missing. Provide previous studies conducted within the health sector using these technologies.
Reviewer 2 Report
Comments and Suggestions for Authors The innovation of the paper is not clear, it should be a technique, which is how to establish the ‘Container-based platform deployment system for real-time multi-user action sharing in virtual reality’. This paper uses a lot of words to describe existing tools and technique such as Docker, Jenkins, VR, etc., but little research results have been found. It seems that this system is only using Docker containers, Jenkins and other tools, combined with VR technique, to apply to neonatal nursing education. This is only the application of existing methods and lacks innovation. The figures in this paper are too simple to understand its meaning clearly.Author Response
Please see the attachment.

Round 2
Reviewer 2 Report
Comments and Suggestions for Authors
1.The Introduction does not provide a good summary of current research, such as how to develop similar platforms using VR technology in similar fields, which leads to unclear innovation points in this paper.
2.There is some overlap between the Results and Discussion sections, and the differences in their content should be clearly defined.
3.No explaination before using the abbreviation "CI/CD"
4.The abbreviation "Spec" is used in the table but not explained
Author Response
Thank you for your comments. We have added a section in the introduction about the research on nursing VR simulation to emphasize the need for this paper. We have also removed redundancies in the results and discussion sections, rewritten them to clarify the differences between each section, and identified all abbreviations used throughout the paper, including “CI/CD” and “Spec”, and organized them into sections.